# Use of soil spectral reflectance to estimate texture and fertility affected by land management practices in Ethiopian tropical highland

**Gizachew Ayalew Tiruneh**[1¤]*, **Derege Tsegaye Meshesha**[2], **Enyew Adgo**[2], **Atsushi Tsunekawa**[3], **Nigussie Haregeweyn**[4], **Ayele Almaw Fenta**[3], **Anteneh Wubet Belay**[2], **Nigus Tadesse**[2], **Genetu Fekadu**[2], **José Miguel Reichert**[5]

**1** Faculty of Agriculture and Environmental Sciences, Debre Tabor University, Debre Tabor, Ethiopia, **2** College of Agriculture and Environmental Sciences, Bahir Dar University, Bahir Dar, Ethiopia, **3** Arid Land Research Center, Tottori University, Tottori, Japan, **4** International Platform for Dryland Research and Education, Tottori University, Tottori, Japan, **5** Soils Department, Universidade Federal de Santa Maria (UFSM), Santa Maria, RS, Brazil

¤ Current address: Department of Natural Resources Management, College of Agriculture and Environmental Sciences, Bahir Dar University, Bahir Dar, Ethiopia
* tiruneh1972@gmail.com

**Data Availability Statement:** Data required for this study are within the paper and/or supplementary files.

## Abstract

As classical soil analysis is time-consuming and expensive, there is a growing demand for visible, near-infrared, and short-wave infrared (Vis-NIR-SWIR, wavelength 350–2500 nm) spectroscopy to predict soil properties. The objectives of this study were to investigate the effects of soil bunds on key soil properties and to develop regression models based on the Vis-NIR-SWIR spectral reflectance of soils in Aba Gerima, Ethiopia. Soil samples were collected from the 0–30 cm soil layer in 48 experimental teff (*Eragrostis tef*) plots and analysed for soil texture, pH, organic carbon (OC), total nitrogen (TN), available phosphorus (av. P), and potassium (av. K). We measured reflectance from air-dried, ground, and sieved soils with a FieldSpec 4 Spectroradiometer. We used regression models to identify and predict soil properties, as assessed by the coefficient of determination ($R^2$), root mean square error (*RMSE*), *bias*, and ratio of performance to deviation (*RPD*). The results showed high variability (CV $\geq$ 35%) and substantial variation ($P < 0.05$ to $P < 0.001$) in soil texture, OC, and av. P in the catchment. Soil reflectance was lower from bunded plots. The pre-processing techniques, including multiplicative scatter correction, median filter, and Gaussian filter for OC, clay, and sand, respectively were used to transform the soil reflectance. Statistical results were: $R^2 = 0.71$, *RPD* = 8.13 and *bias* = 0.12 for OC; $R^2 = 0.93$, *RPD* = 2.21, *bias* = 0.94 for clay; and $R^2 = 0.85$ with *RPD* = 7.54 and *bias* = 0.0.31 for sand with validation dataset. However, care is essential before applying the models to other regions. In conclusion, the findings of this study suggest spectroradiometry can supplement classical soil analysis. However, more research is needed to increase the prediction performance of Vis-NIR-SWIR reflectance spectroscopy to advance soil management interventions.

**Funding:** The research was funded by the Science and Technology Research Partnership for Sustainable Development (grant number JPMJSA1601), Japan Science and Technology Agency/Japan International Cooperation Agency (JICA). Gizachew Ayalew received the fund award. The funders had no role in study design, data collection and analysis, decision to publish, or preparation of the manuscript.

**Competing interests:** The authors have declared that no competing interests exist.

# Introduction

Ethiopia has a wide spatial diversity of soil properties [1]. Improved sustainable land management (SLM) practices including soil bunding may alter key soil qualities [2,3] and control soil loss [4,5], and thereby soil functions [6,7]. SLM activities influence soil physicochemical properties and minimize soil degradation while improving yields [8,9] and they improve people's livelihoods [10]. However, livelihoods are at risk because of soil fertility depletion [11]. Hence, increasing crop production to feed the population is a challenge.

Understanding soil quality requires characterization and assessing spatial variation of soil properties related to crop growth and development [12,13]. Information on soil fertility is needed for site-specific crop and fertilizer management [14,15]. Crop production and yield, plant indicators, and soil texture and colour are widely used to measure soil fertility [16]. The lack of an up-to-date soil database impedes government efforts to boost agricultural production in Ethiopia. Most soil guidelines are not site-specific and soil fertility interventions tend to be blanket recommendations [17,18]. Failing to use soil data can result in nutrient depletion [19], compaction, flooding, and low crop yields [20]. Thus, detailed soil data are essential.

Knowledge of soil resources is essential for developing effective land-use planning and implementing SLM practices. Little information is available for the Upper Blue Nile Basin of Ethiopia, where soils and land-use types patterns fluctuate within small distances [21,22]. Reasons include unavailability and inflated costs of soil laboratory tests, particularly for many soil samples gathered over time and in large areas.

To determine soil properties, Ethiopia has relied on costly and time-consuming traditional analytical methods. However, it faces a crucial difficulty in using these methods owing to the high cost of chemicals and the poor performance of laboratories. As a result, rapid sampling and analysis of soil properties in the field and laboratory are unavailable.

Laboratory soil spectrometry in the visible, near-infrared, and short-wave infrared (Vis-NIR-SWIR, 350–2500 nm) range provides an option for physical and chemical soil studies [23]. Contents of soil OC [24,25], clay [26], TN [27], and soil texture [28] have been well estimated from Vis-NIR-SWIR spectra.

Land-use and management strategies highly influence soil properties [8,29,30]. Understanding variations in soil properties across fields is critical for assessing crop growth and development restrictions related to soil nutrients and to proposing corrective steps for optimal development and effective land-use management [8]. SLM activities, including soil bunds, should address increased human demands and maintain environmental sustainability [31]. Bunds are slope-side embankments made of soil, stone, or a combination of the two. Soil eroded between two bunds is dumped behind the lower bund, which is then lifted to create a bench terrace is formed [32]. Bunds improve soil fertility by reducing runoff and soil loss [33,34].

Furthermore, implementing effective soil bunding can be beneficial in restoring degraded soil quality and functions while ensuring sustainable production. Consequently, understanding how soil properties change as a result of various land management practices is vital to proposing optimal management practices. This research would also help farmers and local planners in developing successful land conservation strategies.

Soil erosion and depleted soil productivity have been major issues in the Aba Gerima catchment, resulting in low crop yields. It would also be useful to replace traditional laboratory soil analysis with more accurate and low-cost methods. Spectroradiometry performed well in measuring soil carbon in different agroecologies, soil forms, and land management activities of Ethiopia [34], but spectroradiometric evidence of the effects of soil bunding on soil characteristics is lacking in the catchment, Ethiopia, and Africa as a whole. Thus, the objectives of this

study were to develop regression models from Vis-NIR-SWIR spectral reflectance data and assess the effects of soil bunding on soil texture, pH, OC, TN, av. P, and av. K by using spectro-radiometric evidence.

## Materials and methods

### Description of the study area

The study was carried out at Aba Gerima (11˚39′0″N– 11˚40′30″N and 37˚29′30″E– 37˚31′30″E), a tropical highland of Ethiopia (Fig 1B). By the Köppen–Geiger classification, the site represents midland, [35], with altitudes varying from 1,900 to 2,000 m above sea level.

Records from 1994 to 2021 at nearby meteorological stations show that the study area receives an average annual rainfall of 1,076 to 1,953 mm and has an average monthly maximum temperature of 27.0˚C and an average monthly minimum temperature of 12.6˚C (Fig 2; S1 Table). The main rainfall occurs from June to August, and the rest of the year is dry [36].

## Methods

### Soil sampling approaches

Cultivated lands were considered in 2019 according to land-use/land-cover information [37]. The catchment having an area of 426 hectare was categorized into three topographic classes, namely gently (2%–5%), moderately (5%–10%), and strongly sloping (10%–15%). There were 24 plots without soil bunds (WB) and 24 plots with soil bunds (SB) reinforced with grass and stone. The soil bunds were built all over the catchment [38,39]. The stone bunds and soil bunds are commonly used on steep slopes (Fig 3A) and moderate slopes (Fig 3B), respectively. The bunds are five years old, have bottom width of 0.8 m and a height of 0.5 m, as described by [40]. Bahir Dar University permitted the work, field site access, and soil sampling in the catchment. Forty-eight representative soil-sampling plots were identified, with a minimum size of 40 m × 40 m (1600 m$^2$) each. All plots were intentionally distributed (Fig 1E). Each plot was geo-referenced with a handheld GPS device (GPSMAP64, Garmin, Olathe, Kansas, USA). In each plot, five soil samples were collected at the top (0–30 cm depth) with an Edelman auger and mixed well in a bucket to make a 1-kg composite soil sample.

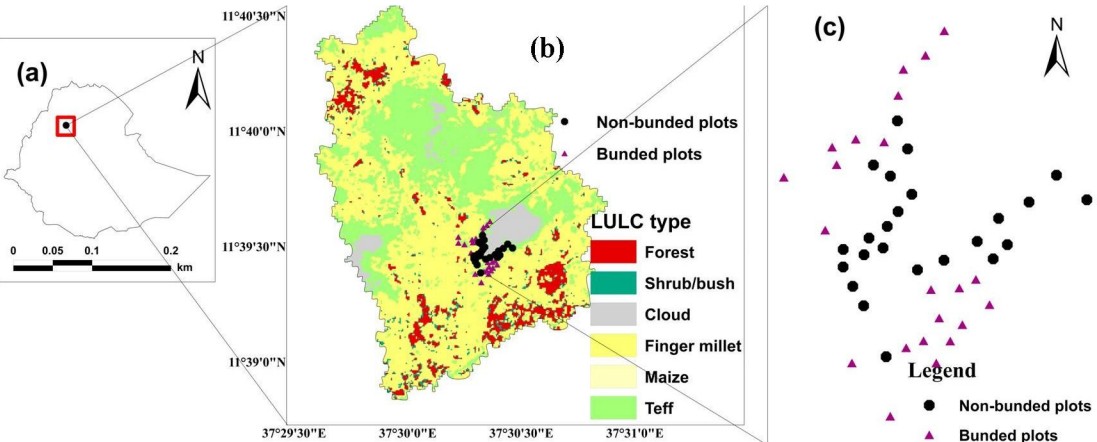

**Fig 1.** Locations of study catchment and soil sampling plots: (a) Ethiopia, (b) Aba Gerima land-use/land-cover (LULC) types, and (c) Soil sampling plots.

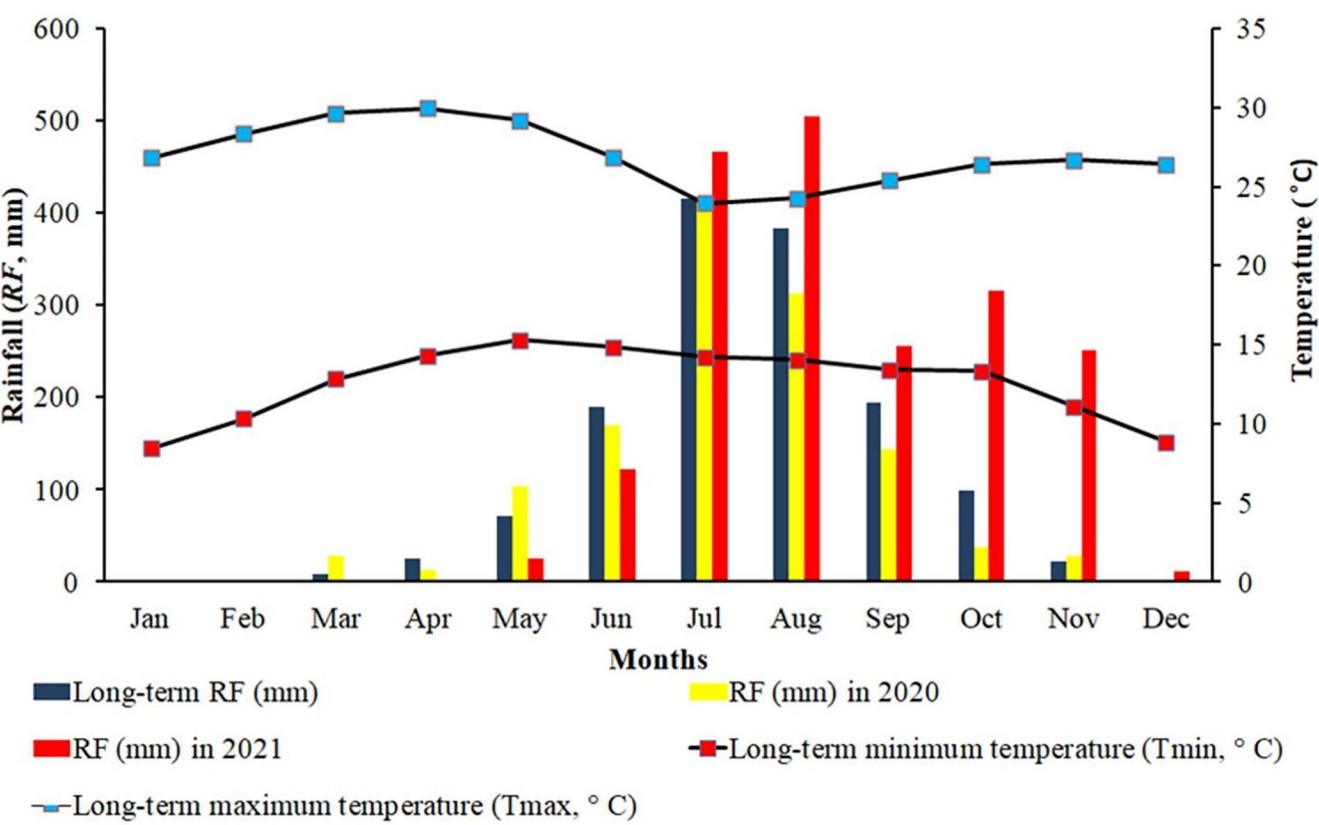

**Fig 2. Long-term (1994–2021) monthly rainfall (RF), maximum temperature ($T_{max}$), and minimum temperature ($T_{min}$) in the study area.**

## Physicochemical soil analysis

Composite soil samples ($n = 48$) were air-dried, ground, and sieved to 2 mm. Later, they were analysed for soil texture, pH, OC, TN, av. P, and av. K at Amhara Design and Supervision Works Enterprise. After the annihilation of organic matter (OM) and soil dispersion, soil texture (sand, silt, and clay) was determined by the hydrometer method [41]. Silt and clay were determined from hydrometer readings after 40 s and clay particles in suspension after 2 h, and the percentage of sand was estimated subtracting by 100 from (clay (%) + sand (%)). Finally, we calculated soil textural classes using the textural triangle of the USDA system [42,43].

Soil pH was determined potentiometrically with a digital pH meter in a 1:2.5 (soil:water) supernatant suspension [44]. Into 100-mL beakers, we poured 10 g of air-dried soil and 25 mL of purified water, stirred it for 1 min with a glass rod and allowed it to equilibrate for 1 h before we measured the pH of the supernatant. Soil OC content was analysed with the wet digestion method, which entails digesting the OC with potassium dichromate in a sulphuric acid solution [45].

Soil TN was determined by the Kjeldahl process, which involves oxidizing OM with condensed sulfuric acid and converting the organic N into ammonia. We weighed 1 g of air-dried soil (<0.5 mm sieve) into a digestion tube; added 2 g of catalyst mixture and a couple of carborundum boiling stones, then stirred the mixture; added 7 mL of concentrated $H_2SO_4$; and digested the mixture on a block digester preheated to 300˚C until the digest was white. After it cooled, we added 50 mL of distilled water, transferred the digest into macro Kjeldahl flasks, and rinsed it with distilled water. We weighed 20 mL of boric acid into a receiver flask, added two drops of indicator solution, and placed the flask under the condenser. Then 75 mL of 40%

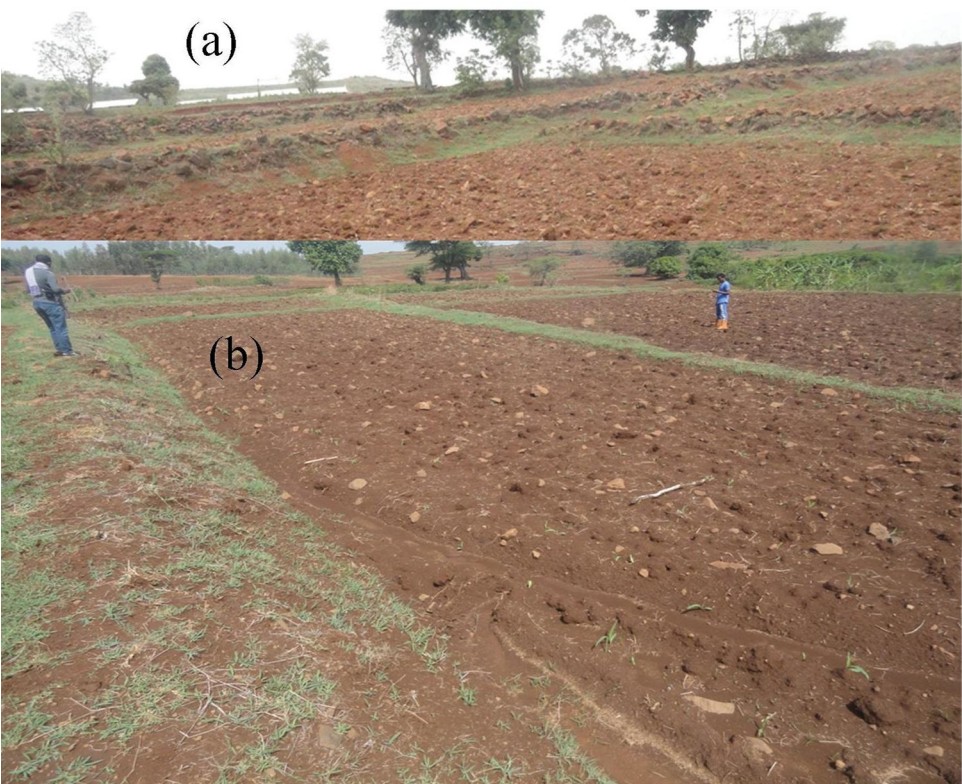

**Fig 3.** Treated with stone-faced soil bunds (a) and with soil bunds (b) at the upper and lower slopes in Aba Gerima catchment.

NaOH was carefully squeezed down the neck of each distillation flask containing the digest, and the mixture was gently stirred. The digests were placed in Kjeldahl distillation flasks, fitted to the appropriate holders, and heated to begin the distillation. The receiver flask was removed when about 80 mL of distillate was obtained. The solution in the receiver flask was stirred with a magnetic stirrer bar and titrated with 0.1 N $H_2SO_4$ from green to pink.

Available P was quantified by the Bray II approach, shaking the soil sample with 0.3 N ammonium fluoride in 0.1 N hydrochloric acid, as described by [46]. The av. P was then measured by spectrophotometer [47]. Available K was analysed by extracting the soil sample with Morgan's solution and measured with a flame photometer [48].

### Soil spectra collection and pre-processing

Each air-dried and ground soil sample (Fig 4D) was placed on a table covered with black geo-membrane. The soil reflectance data in the Vis-NIR-SWIR (350–2500 nm) range were collected with an ASD FieldSpec 4 spectroradiometer (Analytical Spectral Devices [ASD] Inc., Boulder, CO, USA; Fig 4D). Reflectance was measured between 10:30 and 11:00 in direct sunlight. The field of view was set at 25˚, and the distance between the trigger of the spectroradiometer's fibre optic cable and the soil specimen was held at 10 cm for all observations. The spectroradiometer was recalibrated against a white Spectralon (Labsphere Inc., North Sutton, NH, USA) every 10 min.

The radiometer was placed on a table at 1 m above the ground. To avoid unwanted scattering, the table was covered with black geo-membrane. Each scan took 22 s, and a reading was done on each sample. The reflected spectra were recorded and processed in Remote Sensing 3 v.6.4, View Spec Pro v.6.2 software (ASD Inc.). The pre-processing techniques, such as

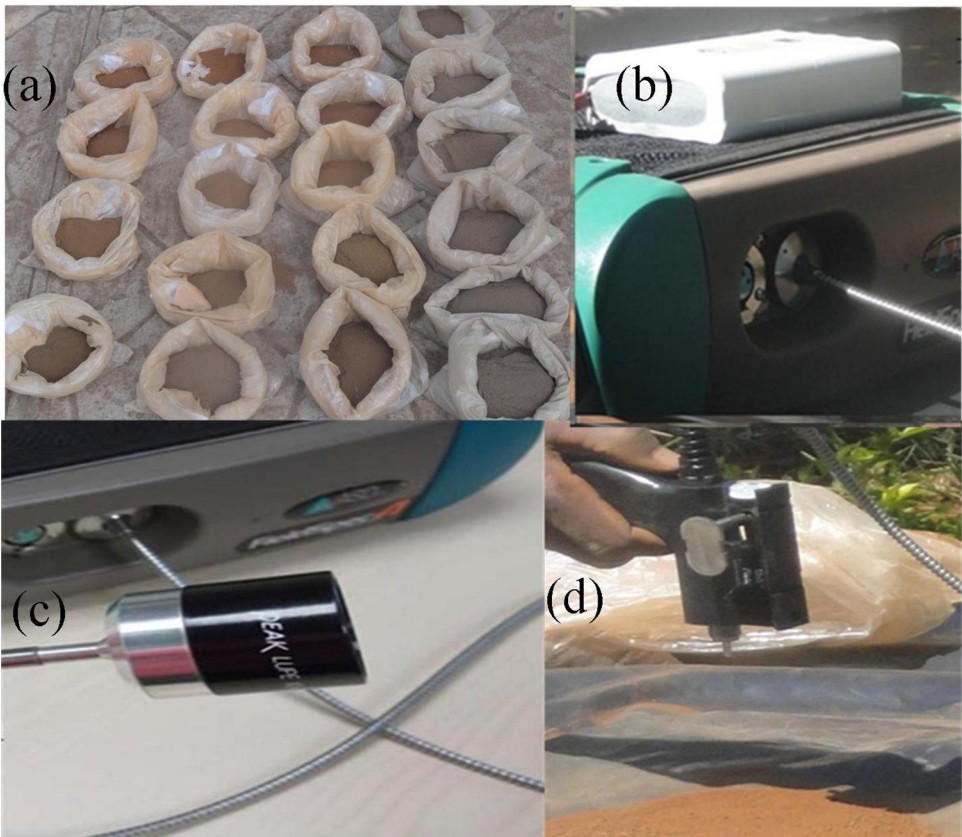

**Fig 4.** Soil spectral measurement: (a) soil samples, (b) FieldSpec Pro spectroradiometer with battery, (c) a close-up of fiber optic cable, and (d) soil reflectance measurement with the sensor.

multiplicative scatter correction, median filter, and Gaussian filter for OC, clay, and sand, respectively were used to transform the spectrum using licensed Unscrambler v.10.5 software (CAMO, Inc., Oslo, Norway). The analysis omitted the noisy spectral regions between 1,340–1,459 nm 1,802–1,971 nm, and 2402–2500 nm [49,50] before spectral modeling.

## Data analysis

Before analysis, we verified the soil dataset ($n = 48$) for normality assumption by the skewness, kurtosis, and Shapiro–Wilk tests ($P < 0.05$). The data were tested by Pearson's linear correlation analysis in SAS v. 9.4 and SPSS v. 24.0 (SPSS Inc., Chicago, IL, USA) software.

We analysed variance and regression in SAS v. 9.4 and separated the means by the least significant difference test ($P < 0.05$). The datasets were divided randomly into 65% of datasets for calibration (31 samples) and 35% of datasets for validation (17 samples). We calculated the coefficient of determination ($R^2$; [51]), root mean square error ($RMSE$), *bias*, and ratio of performance to deviation ($RPD$) [52] (McDowell et al., 2012).) for calibration (31 soil samples) and validation (17 soil samples) as:

$$R^2 = \frac{\sum_{i=1}^{n} (\hat{\mathbf{y}}_i - \bar{\mathbf{y}}_i)^2}{\sum_{i=1}^{n} (y_i - \bar{\mathbf{y}}_i)^2} \qquad \text{Eq1}$$

$$RMSE = \sqrt{\frac{\sum_{i=1}^{n}(\hat{y}_i - y_i)^2}{n}}$$ 
Eq2

$$Bias = \frac{\sum_{1}^{n}(\hat{y}_i - y_i)}{n}$$ 
Eq3

$$RPD = \frac{SD}{\sqrt{\frac{1}{n}\sum_{i=1}^{n}(\hat{y}_{i-}y_i)^2}}$$ 
Eq4

where $n$ is the number of observations, $\hat{y}$ is the predicted value, is the mean observed value, $y$ is the observed (measured) value, and SD is the standard deviation of observed values.

When $R^2$ approaches 1, RPD > 2 RMSE and bias approaches 0, a model improves as a predictor and becomes more efficient [53]. According to [54], models with RPD > 2 are considered "excellent" models with RPD ranging from 1.4 to 2 are considered "acceptable," and models with RPD < 1.4 are considered "poor."

## Results and discussion

### General statistics for measured soil properties

Soil OC, TN, and av. P had high variability (CV ≥ 35%), but soil pH had low variability (CV ≤ 15%) ([55]; Table 1). The high variability in soil properties might be due to soil erosion [56]. The low variability of soil pH corroborates findings of pH varies slightly [6].

The skewness, kurtosis, and Shapiro–Wilk ($P < 0.05$) tests indicated that soil texture, pH, and av. K (Table 1; S1–S3 Figs) met the assumption of homogeneity of variance [57]. However, soil OC, TN, and av. P tended to be logarithmically distributed owing to their positive skewness and slightly asymmetrical distribution (Figs 5 and S1). Similar results are presented in the literature [58].

**Table 1. Descriptive statistics and Shapiro–Wilk probability test of soil parameters in Aba Gerima catchment, Blue Nile basin.**

| Statistic<br>Soil parameter | Mean (μ) ± SEM | SD (σ) | CV (%) | Minimum | Maximum | Skewness | Kurtosis | Shapiro–Wilk<br>P-value |
|---|---|---|---|---|---|---|---|---|
| pH | 5.6 ± 0.04 | 0.25 | 4.52 | 4.7 | 6.01 | −1 | 1.94 | 0.01 |
| Sand (%) | 27.6 ± 1.97 | 13.6 | 49.28 | 5 | 66 | 0.88 | 0.74 | 0.03 |
| Silt (%) | 28.2 ± 1.03 | 7.11 | 25.21 | 13 | 41 | −0.44 | −0.48 | 0.12 |
| Clay (%) | 44.3 ± 2.56 | 17.7 | 39.95 | 11 | 80 | −0.05 | −0.74 | 0.43 |
| OC (%) | 1.65 ± 0.13 | 0.92 | 55.52 | 0.479 | 4.9 | 1.51 | 2.75 | <0.001 |
| log OC (%) | 0.161 ± 0.03 | 0.23 | 139.75 | −0.32 | 0.69 | 0.1 | −0.16 | 0.99 |
| TN (%) | 0.15 ± 0.01 | 0.08 | 49.87 | 0.05 | 0.44 | 1.54 | 3.36 | <0.001 |
| log TN (%) | −0.86 ± 0.03 | 0.2 | −23.49 | −1.3 | −0.36 | 0.12 | −0.04 | 0.98 |
| Av. P (ppm) | 12.2 ± 1.11 | 7.67 | 62.87 | 4.07 | 38 | 1.98 | 3.92 | <0.001 |
| Log Av. P (ppm) | 1.02 ± 0.03 | 0.23 | 22.25 | 0.61 | 1.58 | 0.59 | 0.31 | 0.08 |
| Av. K (g/kg) | 104 ± 3.78 | 26.2 | 25.19 | 41.8 | 150 | −0.51 | 0.03 | 0.14 |

CV (%) = σ/μ × 100, where CV = coefficient of variation; σ = standard deviation (SD), μ = mean; SEM, standard error of the mean; OC, organic carbon; TN, total nitrogen; Log, logarithmic; av. P, available phosphorus; av. K, available potassium.

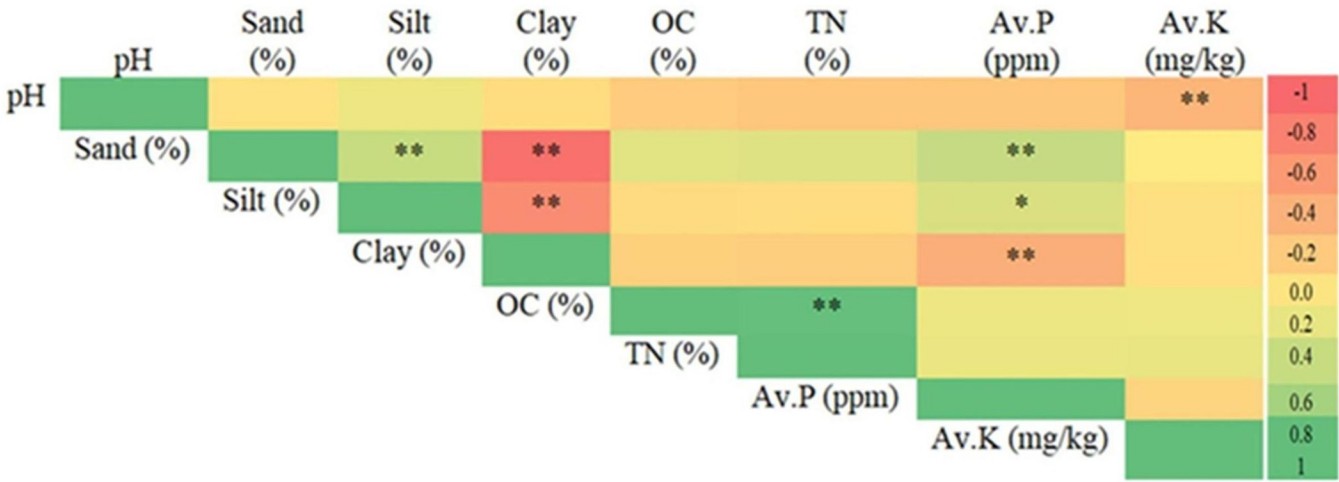

**Fig 5. Pearson's correlation matrix among soil variables in the 0–30 cm soil layer.** Correlation values are colour-coded. OC, organic carbon; TN, total nitrogen; av. P, available phosphorus; av. K, available potassium. Asterisks indicate significant differences: $^*P < 0.05$ and $^{**}P < 0.01$.

### Correlation and regression analyses

Soil pH exhibited a weak relationship with av. K and clay content with av. P (Fig 4; [59]). Soil OC had no relation with av. P or av. K. Similarly (except pH vs. av. K), soil pH and OC did not correlate with K or P contents in Ethiopia [60], Morocco [61], and China [62]. However, the increased soil OC, pH, and clay content might contribute to higher av. P and av. K contents. As the strong positive correlation between soil OC and TN indicated collinearity, we eliminated TN from the variance and spectral analyses. In contrast, [63] found a close relationship between soil OC and TN, as most of the TN was bound in the soil OM.

### Influences of soil bunding on changes in soil properties

Soil texture, OC, and av. P contents varied substantially ($P < 0.05$ to $P < 0.001$) between bunded and non-bunded plots in the three slope classes (Table 2). Soil pH in the Aba Gerima

**Table 2. Analysis of variance in the effect of bunding on soil properties in Aba Gerima.**

| Treatments | Soil parameters | pH | Sand (%) | Silt (%) | Clay (%) | OC (%) | Av. P (ppm) | Av. K (mg kg⁻¹) |
|---|---|---|---|---|---|---|---|---|
| Bunded plots (n = 24) | S1B | 5.62[a] | 18.75[c] | 20.25[c] | 61.00[b] | 2.30[b] | 13.72[b] | 116.55[a] |
| | S2B | 5.72[a] | 22.38[ac] | 31.00[ab] | 46.63[ab] | 1.51[ab] | 10.23[ab] | 97.35[a] |
| | S3B | 5.60[a] | 18.88[c] | 25.75[ac] | 55.38[b] | 1.57[ab] | 11.03[b] | 106.71[a] |
| Non-bunded plots (n = 24) | S1W | 5.54[a] | 33.25[ab] | 31.25[ab] | 35.50[ac] | 1.22[ac] | 10.41[b] | 106.73[a] |
| | S2W | 5.63[a] | 35.50[b] | 33.25[b] | 31.25[c] | 1.54[ab] | 13.03[b] | 101.30[a] |
| | S3W | 5.52[a] | 36.75[b] | 27.50[ab] | 35.75[ac] | 0.90[c] | 6.37[a] | 94.61[a] |
| Mean | | 5.61 | 27.58 | 28.17 | 44.25 | 1.45 | 10.5 | 103.88 |
| CV (%) | | 4.61 | 42.84 | 21.05 | 32.98 | 123.4 | 20.6 | 25.65 |
| LSD | | 0.26 | 11.92 | 5.98 | 14.73 | 0.2 | 0.21 | 26.89 |
| Significance | | ns | ** | *** | *** | *** | * | ns |

OC, soil organic carbon; av. P, available phosphorus; av. K, available potassium. S1, 2%–5%; S2, 5%–10%; S3, 10%–15%; B, soil bund reinforced with stone and grass; W, without soil bund; CV, coefficient of variation; LSD, least significant difference; ns, not significant. n = 48. Values followed by the same letter are not significantly different.

catchment varied from strongly (<5.5) to moderately acidic (5.6–6.5) [64]. In bunded plots, higher soil pH values could be attributed to clay and OM, which retain more basic cations. Lower pH values obtained at non-bunded plots might be due to the inappropriate use of ammonium-based fertilizers and pesticides [65], increased leaching of basic cations, and nitrification ([34,66]. Consequently, the soils of the study area could be affected by acidity problems. Thus, soil pH is a key parameter to monitor the influences of SLM practices on soil quality and crop growth in the region.

The Aba Gerima catchment's soils range from clayey to sandy loam [S4 Fig; 41]. The dominance of silt and clay particles in bunded plots in all three slope classes (S1B, S2B, S3B; Table 2) could be due to the control of soil erosion by bunding. Conversely, the dominance of sand in non-bunded plots on strong slopes (S2W and S3W) might be related to the erosion of finer soil particles [67]. Similarly, bunded soils have higher silt and clay content and lower sand content than non-bunded soils [68,69].

Contents of OC (excluding S1B), av. P, and av. K were low [64], lower than values reported in the same catchment [6] and the Uwite Catchment, Ethiopia [70]. However, the OC concentration of the soil in S1B was within the recommended range for plant growth.

Low levels of OC and av. K could be ascribed to higher rates of erosion due to rainfall, inappropriate cultivation, removal of crop remains and animal dung, a high rate of mineralization through increased temperature, and leaching [71,72]. The highest contents of clay (61%), OC (2.30%), av. P (13.72 ppm), and av. K (116.55 mg/kg) were found in bunded plots on gentle slopes (S1B), possibly due to the accumulation of fine soil particles and available nutrients. These findings support reports that bunding improves soil fertility [6,73] and indicate that it improved clay and silt accumulation, soil OC, av. P, and av. K contents 5 years after implementation.

The pH, OC, av. P, and av. K values were lower than critical levels, explaining reduced soil quality in the Aba Gerima catchment. Thus, enhancing the quality of Ethiopian soils requires increasing soil OM content through the implementation of SLM methods and biomass accumulation [8,74].

## Reflectance characteristics of soils

The mean spectral reflectance of the 48 soil samples tended to increase between 350 and 1100 nm (visible and near-infrared regions) (Fig 6). The lower reflectance of soils from bunded plots could be due to higher soil OM content and smaller particle size [75,76] or to intensive mineral fertilization, higher microbial activity, and lower soil pH [77]. As evidenced by the absorption peaks in Fig 6 [78,79], it could potentially be attributable to the Fe-OH or Mg-OH in the soils.

The soils of the catchment vary from clayey to sandy loam soils (S2 Fig). Reflectance was highest from sandy loam soils throughout the spectrum, lowest from clay soils at 400–1000 nm and loam soils at 1000–2400 nm (Fig 7). As a result, smaller soil particles have higher reflectance. This finding is consistent the results [80] that sandy soils had higher reflectance and clay soils had lower reflectance. As particle size decreases, multiple scattering increases, thus increasing reflectance [81]. Furthermore, the soil spectral reflectance curve showed different trends at different wavelengths, rising rapidly at 400–600 nm and more steadily at 800–2450 nm.

Soils with higher OC content are darker and have lower spectral reflectance than soils with lower OC content (Fig 8) [82]. The presence of OM strongly influences soil reflectance, which decreases as OM content increases [83]. Similarly, as soil moisture increases, the reflectance of incoming visible light falls consistently, making soils look darker [84]. In comparison to dark soils, red soils have less OM and more iron oxides. As a result, soils rich in iron have greater reflectance than soils rich in soil OC [85].

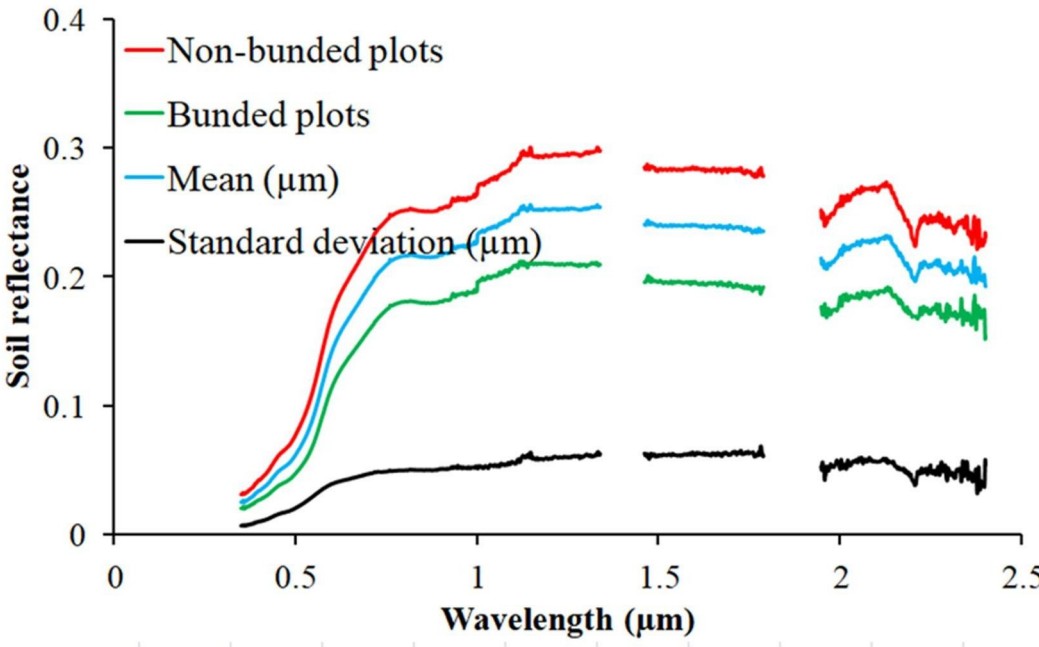

**Fig 6. Soil reflectance spectra with and without bunding.**

## Modeling of soil properties

Fig 9 shows the results of the PLSR analysis calibrated to predict OC, clay, and sand content using pre-processed soil reflectance. For each visible band (400–700 nm), near-infrared band (701–1300 nm), and short-wave infrared band (1301–2500 nm), the red box shows the high loading and influential wavelengths associated with soil properties Higher loading values could indicate which portions of the wavelengths are the most influential in the calibration.

In the visible range, wavelengths with the highest correlation loading values are 355 and 570 nm for organic carbon, and 568 and 570 nm for clay content; 568 and 570 nm for sand content prediction. Thus, the visible spectrum was the most influential in the OC, clay, and sand prediction models. This behavior could owing be to the soil color, which is dominated by free iron oxides [86,87]. The authors [88] also reported 480–600 nm and 720–820 nm as key spectral regions for the OC prediction model.

In the NIR region, the band found at 845 and 850 nm for organic carbon and clay and 1290 nm for sand can be related to the chromophorous components mainly hematite and goethite [89] and the OC content [90]. Furthermore, high correlation loading values were observed at 1592 and 1595 nm for clay and organic carbon content detection at 2293 and 2300 nm. This response could be due to clay, soil water content, and OM content [91,92]. In general, different wavelengths were discovered to be significant for the different soil properties.

As [93] indicated combining the optimum spectral bands was recommended for soil OC detection. The equation for the optimal band combination equations to predict soil OC with the highest $R^2$ (0.92) and the lowest $RMSE$ (0.27) was:

$$OC\,(\%) = 2.47 - 174.45 * \lambda 355\,\text{nm} - 25.21 * \lambda 570\,\text{nm} - 623.61 * \lambda 845\,\text{nm}$$
$$+ 647.64 * \lambda 850\,\text{nm} - 6.89 * \lambda 2293\,\text{nm} + 10.45 * \lambda 2300\,\text{nm}$$

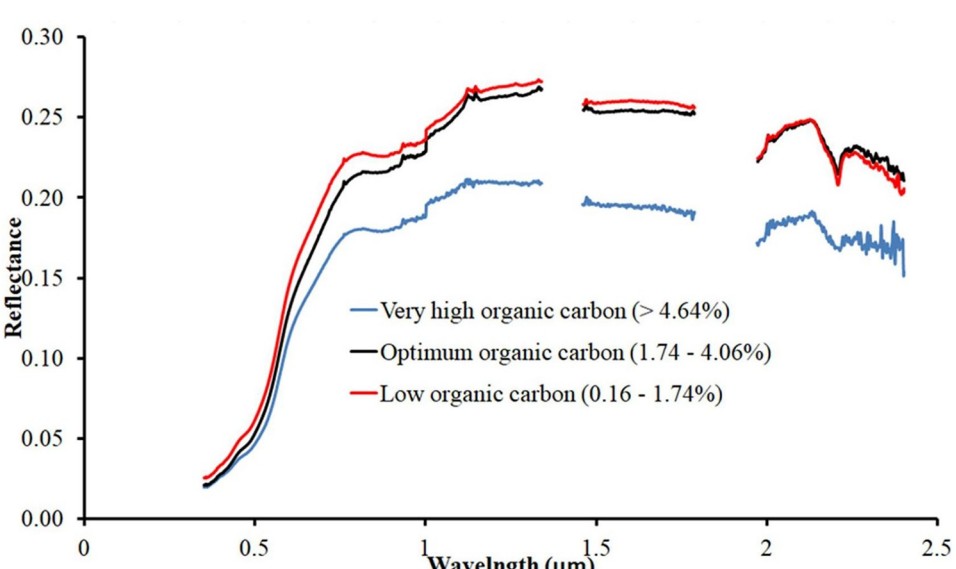

**Fig 7. Influence of soil texture on soil spectral signatures.**

**Fig 8. Effects of soil organic carbon content on soil spectral reflectance.**

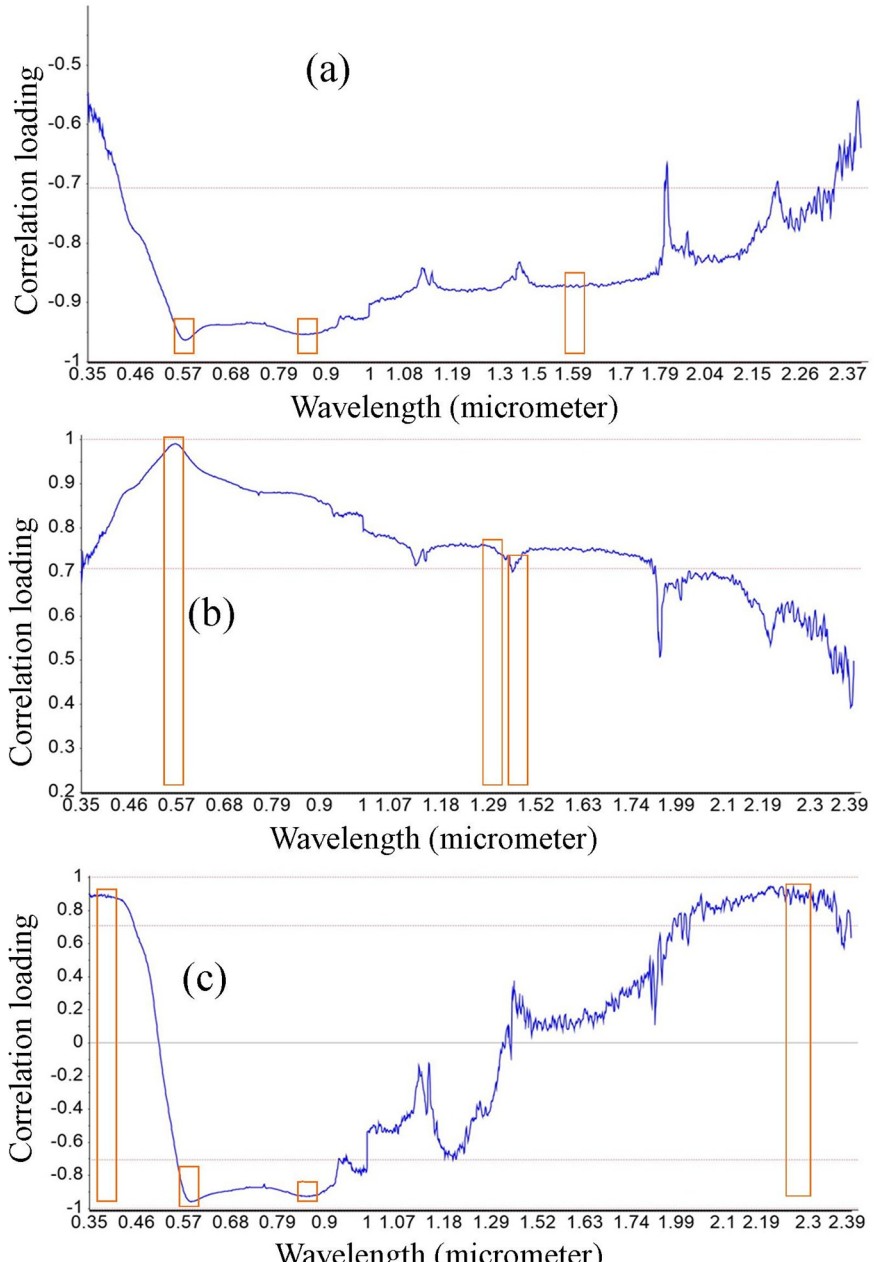

**Fig 9. Loading (contribution) of relevant wavelength for modeling soil properties.** (a) Clay, (b) Sand, and (c) organic carbon contents.

For clay content, the best fit ($R^2$ = 0.86, *RMSE* = 0.60) was:

$$Clay~(\%) = 191.14 - 809.96 * \lambda 568~\text{nm} - 674.25 * \lambda 570~\text{nm} + 2.15 * \lambda 845~\text{nm} \\ - 70.94 * \lambda 850~\text{nm} + 1536.8 * \lambda 1592~\text{nm} - 1421.2 * \lambda 1595$$

For sand, the best fit ($R^2$ = 0.94, *RMSE* = 0.74) was:

$$Sand~(\%) = -63.12 - 628.9 * \lambda 568~\text{nm} + 1477 * \lambda 570~\text{nm} + 1105.7 * \lambda 1290~\text{nm} \\ - 1479.5 * \lambda 1295~\text{nm} + 925.35 * \lambda 1302~\text{nm} - 572.87 \lambda 1305~\text{nm}$$

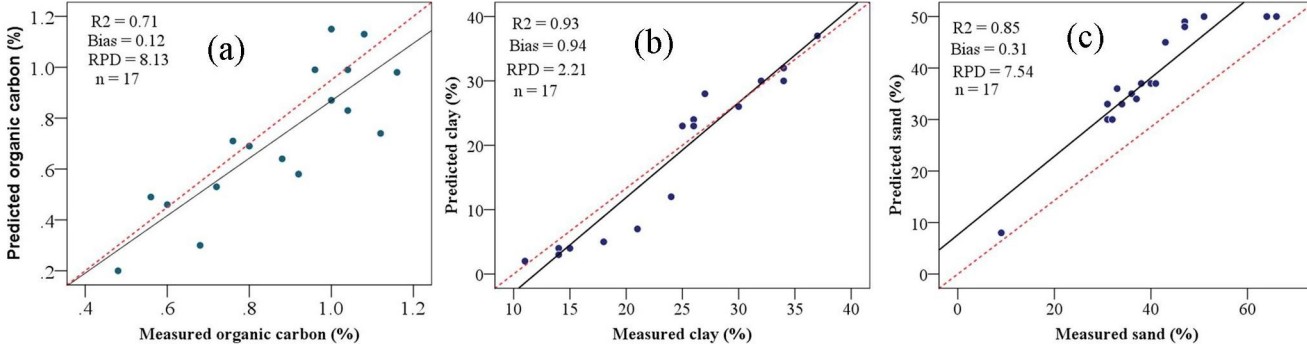

**Fig 10.** Scatterplots of the model validation results for soil organic carbon (a), clay content (b), and sand content (c). $R^2$, coefficient of determination and *RPD*, ratio of performance to deviation.

Based on the regression coefficients, the most influential wavelengths for OC, clay, and sand content prediction were λ850, λ1592, and λ1295 nm, respectively.

We plotted the measured soil properties (%) against the predicted soil properties (%) to validate the model (Fig 10). The simple regression analyses were used the pre-processed dataset (S2 Table) to test the accuracy of prediction of the soil properties. For clay content, the $R^2$, bias, and RPDfor validation dataset were (0.93, 0.94, and 2.21, respectively (Fig 10), better than values reported by 0.70 [94], [95] (0.64), and 0.66 [96], and more accurately than values reported in the literature: $R^2$ = 0.73 with RMSE = 5.40 [28], $R^2$ = 0.83 with RMSE = 0.34 [97], $R^2$ = 0.62 with RMSE = 2.06 [98], and $R^2$ = 0.71–84 [99].

For sand content, we achieved $R^2$ = 0.94 with *RMSE* = 0.74 for calibration and $R^2$ = 0.85 with bias = 0.31 and *RPD* = 7.54 for validation (Fig 10). The sand model was considered excellent according to the $R^2$, *RPD* threshold values [54] (Chang et al. (2001), acceptable bias levels [53] (Bellon-Maurel et al., 2010). Similar values were reported: $R^2$ = 0.80 with RMSE = 3.28 [28], $R^2$ = 0.81 with RMSE = 3.84 [26], $R^2$ = 0.90 with RMSE = 11.66 [98], and $R^2$ = 0.56 to 0.71 [99]. Our predictions of sand content were more accurate than in the literature: $R^2$ = 0.76 with RMSE = 0.92 [97] and $R^2$ = 0.77 [100]. This information could be used to develop and monitor soil management scenarios [26].

For soil OC, we achieved $R^2$ = 0.92 with RMSE = 0.27 for calibration and $R^2$ of 0.71 with bias = 0.12 and RPD = 8.13; bias = 0.12 for validation (Fig 10), which are considered good by $R^2$ threshold values of [101], fair by the *RPD* threshold values [54] (Chang et al. (2001), and acceptable by *bias* values [53].

The OC model's performance is similar to earlier findings: $R^2$ = 0.84–0.93 [102], $R^2$ = 0.63–0.90 with RMSE = 6.40–0.78 [103], $R^2$ = 0.91 [104], $R^2$ = 0.85 with RMSE = 3.77 [105], $R^2$ = 0.57–0.7 [106], $R^2$ = 0.77–0.83 [107,108]; and $R^2$ = 0.764 with RMSE = 0.344 for validation [109,110]. However [111,112], found lower prediction performance ($R^2$ values varying from 0.57 to 0.73 and *RPD* values ranging from 1.80 to 1.93) for soil OC models. In general, regression models based on Vis-NIR-SWIR reflectance spectroscopy could be used to predict soil properties in the study area.

## Conclusions

Soil bunding in the Aba Gerima catchment, Ethiopia, positively influenced clay, silt, OC, and av. K contents of soils. Soil physicochemical properties (texture, OC, and av. P) varied widely between bunded and non-bunded plots in different slope classes. In contrast, soil pH and contents of clay, silt, sand, OC, av. P, and av. K were all higher in soils on lower slopes than on

higher slopes. These findings suggest that site-specific information could inform land management interventions, such as soil bunding, for sustainable soil management. The reflectance of soils from bunded plots was lower, which improved soil fertility.

This study looked at how Vis-NIR-SWIR (350–2500 nm) soil spectral data could be used to determine some soil physical and chemical parameters. The use of regressive functions to estimate the calibrated soil attributes with their pretreatment spectral reflectance data was proposed due to high correlation loading and regression coefficients. Regression models were developed for predicting the clay, sand, and OC contents with acceptable accuracy ($R^2 > 0.70$, $RPD > 2$, and *bias* values $< 0$).

Our findings suggest that Vis-NIR-SWIR reflectance spectroscopy might be utilized for soil characterization, evaluation, and monitoring in a quick and non-destructive manner. The findings have implications for spatial management and monitoring of soil physicochemical properties across the catchment. Soil prediction models based on spectroradiometry will help land-users and policymakers by contributing to the development of sustainable and site-specific soil management strategies.

As the tempo-spatial variation of soil properties between regions would influence the accuracy of the estimation model, caution should be vital before applying the models to other areas. Nonetheless, further research is required to identify which portions of the spectrum contribute to the models, improve the predictive capacity of Vis-NIR-SWIR spectroscopy, support land management interventions, and explore their effects on other soil parameters such as biological properties.

## Supporting information

**S1 Fig. Descriptive summary and Normality test of soil organic carbon (OC), total nitrogen (TN), and available phosphorus (av. P) in Aba Gerima catchment.** SE, standard error; SD, standard deviation; CV = coefficient of variation; min, minimum; max, maximum.
(JPG)

**S2 Fig. Descriptive summary and Normality test of soil pH, sand (%), silt (%), and clay (%) in Aba Gerima catchment.** SE, standard error; SD, standard deviation; CV = coefficient of variation; min, minimum; max, maximum.
(JPG)

**S3 Fig. Descriptive summary and Normality test of logOC (%), logTN (%), logav.P (ppm), and av.K (mg/Kg) in Aba Gerima catchment.** Log, logarithmically transformed; OC, organic carbon; TN, total nitrogen; av. K, available phosphorus; available potassium, min, minimum and max, maximum.
(JPG)

**S4 Fig. Soil textural classes in Aba Gerima catchment.**
(JPG)

**S1 Table. Climatic data of Aba Gerima catchment.**
(XLSX)

**S2 Table. Pre-processed soil spectral data for selected wavelengths of study catchment.**
(XLSX)

## Acknowledgments

We thank Agerselam Gualie and Melkamu Wudu for their support during our field and laboratory work. The authors acknowledged the Japan International Cooperation Agency (JICA)

for purchasing an ASD FieldSpec 4 spectroradiometer and the licensed Unscrambler software for the study. The authors also express their gratitude to the reviewers and editors for their insightful comments and recommendations.

## Author Contributions

**Conceptualization:** Gizachew Ayalew Tiruneh, Derege Tsegaye Meshesha, Enyew Adgo, Nigussie Haregeweyn, Ayele Almaw Fenta, Nigus Tadesse, Genetu Fekadu, José Miguel Reichert.

**Data curation:** Gizachew Ayalew Tiruneh, Derege Tsegaye Meshesha, Nigus Tadesse, Genetu Fekadu, José Miguel Reichert.

**Formal analysis:** Ayele Almaw Fenta, Nigus Tadesse, Genetu Fekadu, José Miguel Reichert.

**Funding acquisition:** Atsushi Tsunekawa, Nigussie Haregeweyn.

**Investigation:** Anteneh Wubet Belay, José Miguel Reichert.

**Methodology:** Gizachew Ayalew Tiruneh, Derege Tsegaye Meshesha, Ayele Almaw Fenta.

**Project administration:** Atsushi Tsunekawa, Ayele Almaw Fenta, Anteneh Wubet Belay.

**Supervision:** Gizachew Ayalew Tiruneh, Derege Tsegaye Meshesha, Enyew Adgo, Nigussie Haregeweyn, Ayele Almaw Fenta.

**Validation:** Gizachew Ayalew Tiruneh, Derege Tsegaye Meshesha, Enyew Adgo, Nigussie Haregeweyn, Nigus Tadesse, Genetu Fekadu.

**Writing – original draft:** Gizachew Ayalew Tiruneh, Derege Tsegaye Meshesha.

**Writing – review & editing:** Gizachew Ayalew Tiruneh, Derege Tsegaye Meshesha, Enyew Adgo, Atsushi Tsunekawa, Nigussie Haregeweyn, Ayele Almaw Fenta, Anteneh Wubet Belay, Nigus Tadesse, Genetu Fekadu, José Miguel Reichert.

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
