## [Decision Letter · Decision Letter 0]

28 Apr 2022

PONE-D-22-07890Use of soil spectral reflectance to estimate texture and fertility affected by land management practices in Ethiopian tropical highlandPLOS ONE

Dear Dr. Tiruneh,

Thank you for submitting your manuscript to PLOS ONE. After careful consideration, we feel that it has merit but does not fully meet PLOS ONE’s publication criteria as it currently stands. Therefore, we invite you to submit a revised version of the manuscript that addresses the points raised during the review process.

We look forward to receiving your revised manuscript.

Kind regards,

Chun Liu

Academic Editor

PLOS ONE

Journal Requirements:

Whilst you may use any professional scientific editing service of your choice, PLOS has partnered with both American Journal Experts (AJE) and Editage to provide discounted services to PLOS authors. Both organizations have experience helping authors meet PLOS guidelines and can provide language editing, translation, manuscript formatting, and figure formatting to ensure your manuscript meets our submission guidelines. To take advantage of our partnership with AJE, visit the AJE website (http://aje.com/go/plos) for a 15% discount off AJE services. To take advantage of our partnership with Editage, visit the Editage website (www.editage.com) and enter referral code PLOSEDIT for a 15% discount off Editage services.  If the PLOS editorial team finds any language issues in text that either AJE or Editage has edited, the service provider will re-edit the text for free.

A clean copy of the edited manuscript (uploaded as the new *manuscript* file).

4. We note that Figure 1 in your submission contain map/satellite images which may be copyrighted. All PLOS content is published under the Creative Commons Attribution License (CC BY 4.0), which means that the manuscript, images, and Supporting Information files will be freely available online, and any third party is permitted to access, download, copy, distribute, and use these materials in any way, even commercially, with proper attribution. For these reasons, we cannot publish previously copyrighted maps or satellite images created using proprietary data, such as Google software (Google Maps, Street View, and Earth). For more information, see our copyright guidelines: http://journals.plos.org/plosone/s/licenses-and-copyright.

a) You may seek permission from the original copyright holder of Figure 1 to publish the content specifically under the CC BY 4.0 license.  

Natural Earth (public domain): http://www.naturalearthdata.com/.

5. Please ensure that you refer to Figures 5, 10 and 11 in your text as, if accepted, production will need this reference to link the reader to the figure.

7.  We note that you have indicated that data from this study are available upon request. PLOS only allows data to be available upon request if there are legal or ethical restrictions on sharing data publicly. For more information on unacceptable data access restrictions, please see http://journals.plos.org/plosone/s/data-availability#loc-unacceptable-data-access-restrictions. 

Reviewers' comments:

Reviewer's Responses to Questions

**Comments to the Author**

1. Is the manuscript technically sound, and do the data support the conclusions?

Reviewer #1: No

Reviewer #2: Partly

2. Has the statistical analysis been performed appropriately and rigorously? 

Reviewer #1: No

Reviewer #2: No

3. Have the authors made all data underlying the findings in their manuscript fully available?

Reviewer #1: No

Reviewer #2: Yes

4. Is the manuscript presented in an intelligible fashion and written in standard English?

Reviewer #1: No

Reviewer #2: No

5. Review Comments to the Author

Reviewer #1: This paper presents the use of Vis-NIR spectrometer to measure soil fertility parameters for some experimental sites in Ethiopia. While interesting, unfortunately the study was conducted on a small number of samples, even if the prediction is meaningful it does not have any meaning or scientific contribution. The relationship between spectra index and soil properties are dubious and there is no independent validation of the data. As such, this paper has serious flaws in the analysis and provided no real value.

Reviewer #2: The authors are presenting an interesting research topic addressing current needs for understanding soil fertility status which play important role in addressing food security and climate change challenges. The choice of technique to analyze soil is laudable especially when large areas are targeted.

However, the authors fell to explain the end-to-end scientific workflow for the procedure. All spectroscopic datasets are known to contain some level of redundant information in form of noise which should be filtered prior to modeling. This step was not explained.

Another important lacking is how the simple linear regression models were created. Which spectral bands were used? How were they selected?

The choice of Nash- Sutcliffe Efficiency is mostly used in simulation models. The authors should justify why it was used. Other metrics commonly used and acceptable in spectroscopy studies like bias, RPD should be considered.

Closely related to this aspect of metric is about selection of calibration (training set) and validation (testing sets) from the 48 spectra acquired.

I recommend further work to be done on this paper to improve on it is general quality.

I highlight below specific areas which requires amendment.

L127 .. well mixed…should be … mixed well…

L129 correct repeated word ‘at’.

Determination of Sand content from the difference as authors suggests on line 137 should be written clearly.

L173, mean spectra should not be obtained before checking for quality of spectra to screen bad spectra. Authors should outline what measures were taken to avoid including a bad spectrum (‘noisy’ replicate among the five into the averaged).

L180 to 183 – The requirement for assumption of normality is not clear. Why was this necessary? Was this done on the spectral dataset or on the physicochemical data?

L302 to 303 is not clear; it should be rewritten.

L310 to L315 gives model summary statistics for R-squared as a range instead of a single value for the calibration data set. Why is this?

6. PLOS authors have the option to publish the peer review history of their article (what does this mean?). If published, this will include your full peer review and any attached files.

Reviewer #1: No

Reviewer #2: **Yes: **ok

---

## [Author Response · Author response to Decision Letter 0]

18 May 2022

Date: May 17, 2022

Rebuttal letter

PONE-D-22-07890

We are happy about the academic editor and the reviewers’ comments, which strengthen the current version of the manuscript “Use of soil spectral reflectance to estimate texture and fertility affected by land management practices in Ethiopian tropical highland”. In addition, our supreme sincere gratitude goes to you and the reviewers who devote their valuable time to bring our manuscript to a competent paper. 

We have provided a one by one reply to all concerns and comments given below. We thank you for your consideration of this resubmission and look forward to your response.

Best regards,

Gizachew Ayalew Tiruneh (on behalf of all co-authors)

Lecturer in Debre Tabor University

Ph.D. Fellow in soil science, Bahir Dar University

Email: tiruneh1972@gmail.com

RESPONSE TO ALL COMMENTS

Dear editor and reviewers, thank you so much for taking your valuable time to elevate the quality of our manuscript. We do hope that the editor’s and Reviewer’s concerns will be addressed.

Editor comments:

Comment 1: A rebuttal letter that responds to each point raised by the academic editor and reviewer(s). You should upload this letter as a separate file labeled 'Response to Reviewers'.

Response: We addressed the concerns provided by the editor and reviewers and uploaded a file labeled “Response to Reviewers”.

Comment 2: A marked-up copy of your manuscript that highlights changes made to the original version. You should upload this as a separate file labeled 'Revised Manuscript with Track Changes'.

Response: We tried to highlight our revised paper with tracked changes. We uploaded this as a separate file labeled 'Tracked changes'.

Comment 3: An unmarked version of your revised paper without tracked changes. You should upload this as a separate file labeled 'Manuscript'.

Response: We revised our manuscript without tracked changes. We uploaded this as a separate file labeled 'Manuscript'.

Comments 4: Response: We have not made any changes to financial disclosure.

Journal Requirements:

Comment 1: Please ensure that your manuscript meets PLOS ONE's style requirements, including those for file naming. The PLOS ONE style templates can be found at 

Response: Thank you. We tried to follow the PLOS ONE's style requirements throughout the manuscript.

Comment 2: We suggest you thoroughly copyedit your manuscript for language usage, spelling, and grammar. If you do not know anyone who can help you do this, you may wish to consider employing a professional scientific editing service. 

Whilst you may use any professional scientific editing service of your choice, PLOS has partnered with both American Journal Experts (AJE) and Editage to provide discounted services to PLOS authors. Both organizations have experience helping authors meet PLOS guidelines and can provide language editing, translation, manuscript formatting, and figure formatting to ensure your manuscript meets our submission guidelines. To take advantage of our partnership with AJE, visit the AJE website (http://aje.com/go/plos) for a 15% discount off AJE services. To take advantage of our partnership with Editage, visit the Editage website (www.editage.com) and enter referral code PLOSEDIT for a 15% discount off Editage services. If the PLOS editorial team finds any language issues in text that either AJE or Editage has edited, the service provider will re-edit the text for free.

Response: Thank you for your advice. We have thoroughly revised our manuscript with the help of Grammarly (premium), ELSS editing service, and licensed iThenticate software (as attached), and we do hope that the reviewers concerns will be addressed.

Comment 3: Upon resubmission, please provide the following:

The name of the colleague or the details of the professional service that edited your manuscript. A copy of your manuscript showing your changes by either highlighting them or using track changes (uploaded as a *supporting information* file)

A clean copy of the edited manuscript (uploaded as the new *manuscript* file).

Response: The details of the colleague: 

Name Tiringo Yilak Alemayeh, Place: Debre Tabor University, Work: Senior lecturer, and email address: tiringoy4@gmail.com

Name José Miguel Reichert, Place: Universidade Federal de Santa Maria (UFSM), Work: Professor of Soil Science and email address: reichert@ufsm.br

English language editing professional service was also obtained as you refer in the attached document. We tried to highlight our revised paper with tracked changes uploaded as a *supporting information* file. 

Comment 4: In your Methods section, please provide additional information regarding the permits you obtained for the work. Please ensure you have included the full name of the authority that approved the field site access and, if no permits were required, a brief statement explaining why.

Response: Bahir Dar University officially approved the study, field site access, and soil sampling in the Aba Gerima catchment (the details are found from ethics statement from attached document).

Comment 5: We note that Figure 1 in your submission contains map/satellite images which may be copyrighted. All PLOS content is published under the Creative Commons Attribution License (CC BY 4.0), which means that the manuscript, images, and Supporting Information files will be freely available online, and any third party is permitted to access, download, copy, distribute, and use these materials in any way, even commercially, with proper attribution. For these reasons, we cannot publish previously copyrighted maps or satellite images created using proprietary data, such as Google software (Google Maps, Street View, and Earth). For more information, see our copyright guidelines: http://journals.plos.org/plosone/s/licenses-and-copyright.

a) You may seek permission from the original copyright holder of Figure 1 to publish the content specifically under the CC BY 4.0 license. 

Natural Earth (public domain): http://www.naturalearthdata.com/.

Figure 1 study area description and sampling distribution

Response: Thank you for your suggestion. In Response, we replaced the Figure 1 with the new one. 

Comment 6: Please ensure that you refer to Figures 5, 10 and 11 in your text as, if accepted, production will need this reference to link the reader to the figure.

Response: Thank you for your suggestion, and we cited Figure 5 in the text and removed Figures 10 and 11 from the revised manuscript.

Comment 7: Please include captions for your Supporting Information files at the end of your manuscript, and update any in-text citations to match accordingly. Please see our Supporting Information guidelines for more information: http://journals.plos.org/plosone/s/supporting-information. 

Response: Thank you for your advice, and we added Supporting Information files at the end of the revised manuscript with their captions.

Comment 8: We note that you have indicated that data from this study are available upon request. PLOS only allows data to be available upon request if there are legal or ethical restrictions on sharing data publicly. For more information on unacceptable data access restrictions, please see http://journals.plos.org/plosone/s/data-availability#loc-unacceptable-data-access-restrictions. 

Response: Data required for this study are within the manuscript and/or supplementary files.

 

Reviewer comments:

Comment 1. Is the manuscript technically sound, and do the data support the conclusions?

Reviewer #1: No

Reviewer #2: Partly

Response: Thank you. We appreciate your valuable comments. We tried to address the comments and incorporated them in the revised manuscript. We hope that this revised version will be satisfying.

Comment 2. Has the statistical analysis been performed appropriately and rigorously?

 Reviewer #1: No

Reviewer #2: No

Response: Thank you. We have gone thoroughly the revised manuscript, and hopefully that the reviewers will be satisfied.

Comment 3: Have the authors made all data underlying the findings in their manuscript fully available?

 Reviewer #1: No

Reviewer #2: Yes

Response: Thank you. Data required for this study are within the manuscript and/or supplementary files.

Comment 4. Is the manuscript presented in an intelligible fashion and written in standard English?

 Reviewer #1: No

Reviewer #2: No

Response: Thank you for your advice. We have thoroughly revised our manuscript with the help of Grammarly (premium), ELSS editing service, and licensed iThenticate software (as attached documents), and we do hope that the reviewers concerns will be addressed.

5. Review Comments to the Author

Reviewer #1: This paper presents the use of Vis-NIR spectrometer to measure soil fertility parameters for some experimental sites in Ethiopia. While interesting, unfortunately the study was conducted on a small number of samples, even if the prediction is meaningful it does not have any meaning or scientific contribution. The relationship between spectra index and soil properties are dubious and there is no independent validation of the data. As such, this paper has serious flaws in the analysis and provided no real value.

Response: Thank you for your concern and advice. We deleted the issue of relationship between spectral index and soil properties from the revised manuscript.

Reviewer #2: The authors are presenting an interesting research topic addressing current needs for understanding soil fertility status which play important role in addressing food security and climate change challenges. The choice of technique to analyze soil is laudable especially when large areas are targeted.

However, the authors fell to explain the end-to-end scientific workflow for the procedure. All spectroscopic datasets are known to contain some level of redundant information in form of noise which should be filtered prior to modeling. This step was not explained.

Another important lacking is how the simple linear regression models were created. Which spectral bands were used? How were they selected?

The choice of Nash- Sutcliffe Efficiency is mostly used in simulation models. The authors should justify why it was used. Other metrics commonly used and acceptable in spectroscopy studies like bias, RPD should be considered.

Closely related to this aspect of metric is about selection of calibration (training set) and validation (testing sets) from the 48 spectra acquired.

I recommend further work to be done on this paper to improve on it is general quality.

Response: Thank you for your kind suggestions. The pre-processing techniques such as multiplicative scatter correction for OC, median filter for clay, and Gaussian filter for sand were used to reduce “noise” from the raw spectrum before modeling of soil parameters. Besides, redundant spectral information was minimized through selecting wavelengths with high loading values and regression coefficients at each visible, near-infrared, and short-wave infrared spectrum for the studied soil parameter using licensed Unscrambler software. We removed Nash-Sutcliffe Efficiency and newly added metrics such as bias, RPD to validate the independent validation dataset (17 soil samples) from the 48 spectra. We hope that the reviewers concerns will be addressed.

Reviewer #2: I highlight below specific areas which requires amendment.

L127 .. well mixed…should be … mixed well…

L129 correct repeated word ‘at’.

Determination of Sand content from the difference as authors suggests on line 137 should be written clearly.

L173, mean spectra should not be obtained before checking for quality of spectra to screen bad spectra. Authors should outline what measures were taken to avoid including a bad spectrum (‘noisy’ replicate among the five into the averaged).

L180 to 183 – The requirement for assumption of normality is not clear. Why was this necessary? Was this done on the spectral dataset or on the physicochemical data?

L302 to 303 is not clear; it should be rewritten.

L310 to L315 gives model summary statistics for R-squared as a range instead of a single value for the calibration data set. Why is this?

Response: Thank you for the comments. We tried to incorporate the above comments in the revised manuscript. We hope that this revised version will be satisfying.

Comment: While revising your submission, please upload your figure files to the Preflight Analysis and Conversion Engine (PACE) digital diagnostic tool, https://pacev2.apexcovantage.com/. PACE helps ensure that figures meet PLOS requirements. To use PACE, you must first register as a user. Registration is free. Then, login and navigate to the UPLOAD tab, where you will find detailed instructions on how to use the tool. If you encounter any issues or have any questions when using PACE, please email PLOS at figures@plos.org. Please note that Supporting Information files do not need this step.

Response: Thank you. We have used PACE with this submission, so this should be right.

Please note that once again, thank you very much. Your comments are greatly appreciated.

Best regards,

Gizachew Ayalew Tiruneh (on behalf of all co-authors)

Lecturer in Debre Tabor University

Ph.D. Fellow in soil science, Bahir Dar University, Email: tiruneh1972@gmail.com

---

## [Decision Letter · Decision Letter 1]

15 Jun 2022

Use of soil spectral reflectance to estimate texture and fertility affected by land management practices in Ethiopian tropical highland

PONE-D-22-07890R1

Dear Dr. Tiruneh,

We’re pleased to inform you that your manuscript has been judged scientifically suitable for publication and will be formally accepted for publication once it meets all outstanding technical requirements.

Kind regards,

Chun Liu

Academic Editor

PLOS ONE

Additional Editor Comments (optional):

Reviewers' comments:

Reviewer's Responses to Questions

**Comments to the Author**

1. If the authors have adequately addressed your comments raised in a previous round of review and you feel that this manuscript is now acceptable for publication, you may indicate that here to bypass the “Comments to the Author” section, enter your conflict of interest statement in the “Confidential to Editor” section, and submit your "Accept" recommendation.

Reviewer #2: All comments have been addressed

2. Is the manuscript technically sound, and do the data support the conclusions?

Reviewer #2: Yes

3. Has the statistical analysis been performed appropriately and rigorously? 

Reviewer #2: Yes

4. Have the authors made all data underlying the findings in their manuscript fully available?

Reviewer #2: Yes

5. Is the manuscript presented in an intelligible fashion and written in standard English?

Reviewer #2: Yes

6. Review Comments to the Author

Reviewer #2: The authors have addressed review comments adequately.

For future work an approach that does not use linear models should be tested to ensure multicollinearities from highly correlated variables is addressed.

7. PLOS authors have the option to publish the peer review history of their article (what does this mean?). If published, this will include your full peer review and any attached files.

Reviewer #2: **Yes: **Dr. Andrew M. Sila

---

## [Editor Report · Acceptance letter]

12 Jul 2022

PONE-D-22-07890R1 

Use of soil spectral reflectance to estimate texture and fertility affected by land management practices in Ethiopian tropical highland 

Dear Dr. Tiruneh:

I'm pleased to inform you that your manuscript has been deemed suitable for publication in PLOS ONE. Congratulations! Your manuscript is now with our production department. 

Kind regards, 

on behalf of

Dr. Chun Liu 

Academic Editor

PLOS ONE